# A NEW FRAMEWORK FOR TENSOR PCA BASED ON TRACE INVARIANTS

## ABSTRACT

We consider the Principal Component Analysis (PCA) problem for tensors $\mathbf{T} \in (\mathbb{R}^n)^{\otimes k}$ of large dimension $n$ and of arbitrary order $k \geq 3$. It consists in recovering a spike $\boldsymbol{v}_0^{\otimes k}$ (related to a signal vector $\boldsymbol{v}_0 \in \mathbb{R}^n$) corrupted by a Gaussian noise tensor $\mathbf{Z} \in (\mathbb{R}^n)^{\otimes k}$ such that $\mathbf{T} = \beta \boldsymbol{v}_0^{\otimes k} + \mathbf{Z}$ where $\beta$ is the signal-to-noise ratio. In this paper, we propose a new framework based on tools developed by the theoretical physics community to address this important problem. They consist in trace invariants of tensors built by judicious contractions (extension of matrix product) of the indices of the tensor $\mathbf{T}$. Inspired by these tools, we introduce a new process that builds for each invariant a matrix whose top eigenvector is correlated to the signal for $\beta$ sufficiently large. Then, we give examples of classes of invariants for which we demonstrate that this correlation happens above the best algorithmic threshold ($\beta \geq n^{k/4}$) known so far. This method has many algorithmic advantages: (i) it provides a detection algorithm linear in time and with only $O(1)$ memory requirements (ii) the algorithms are very suitable for parallel architectures and have a lot of potential of optimization given the simplicity of the mathematical tools involved (iii) experimental results show an improvement of the state of the art for the symmetric tensor PCA. We provide experimental results to these different cases that match well with our theoretical findings.

## 1 INTRODUCTION

Powerful computers and acquisition devices have made it possible to capture and store real-world multidimensional data. For practical applications (Kolda & Bader (2009)), analyzing and organizing these high dimensional arrays (formally called tensors) lead to the well known curse of dimensionality (Gao et al. (2017),Suzuki (2019)). Thus, dimensionality reduction is frequently employed to transform a high-dimensional data set by projecting it into a lower dimensional space while retaining most of the information and underlying structure. One of these techniques is Principal Component Analysis (PCA), which has made remarkable progress in a large number of areas thanks to its simplicity and adaptability (Jolliffe & Cadima (2016); Seddik et al. (2019)).

In the Tensor PCA, as introduced by Richard & Montanari (2014), we consider a model where we attempt to detect and retrieve an unknown unit vector $\boldsymbol{v}_0$ from noise-corrupted multi-linear measurements put in the form of a tensor $\mathbf{T}$. Using the notations found below, our model consists in:

$$\mathbf{T} = \beta \boldsymbol{v}_0^{\otimes k} + \mathbf{Z}, \tag{1}$$

with $\mathbf{Z}$ a pure Gaussian noise tensor of order k and dimension $n$ with identically independent distributed (iid) standard Gaussian entries: $\mathbf{Z}_{i_1, i_2, \ldots, i_k} \sim \mathcal{N}(0, 1)$ and $\beta$ the signal-to-noise ratio.

To solve this important problem, many methods have been proposed. However, practical applications require optimizable and parallelizable algorithms that are able to avoid the high computationally cost due to an unsatisfactory scalability of some of these methods. A summary of the time and space requirement of some existent methods can be found in Anandkumar et al. (2017).

One way to achieve this parallelizable algorithm is through methods based on tensor contractions (Kim et al. (2018)) which are extensions of the matrix product. These last years, tools based on tensor contractions have been developed by theoretical physicists where random tensors have emerged as a generalization of random matrices. In this paper, we investigate the algorithmic threshold of

tensor PCA and some of its variants using the theoretical physics approach and we show that it leads to new insights and knowledge in tensor PCA.

Tensor PCA and tensor decomposition (the recovery of multiple spikes) is motivated by the increasing number of problems in which it is crucial to exploit the tensorial structure (Sidiropoulos et al. (2017)). Recently it was successfully used to address important problems in unsupervised learning (learning latent variable models, in particular latent Dirichlet allocation Anandkumar et al. (2014), Anandkumar et al. (2015)), supervised learning (training of two-layer neural networks, Janzamin et al. (2015)) and reinforcement learning (Azizzadenesheli et al. (2016)).

**Related work**    Tensor PCA was introduced by Richard & Montanari (2014) where the authors suggested and analyzed different methods to recover the signal vector like matrix unfolding and power iteration. Since then, various other methods were proposed. Hopkins et al. (2015) introduced algorithms based on the sum of squares hierarchy with the first proven algorithmic threshold of $n^{k/4}$. However this class of algorithm generally requires high computing resources and relies on complex mathematical tools (which makes its algorithmic optimization difficult). Other studied methods have been inspired by different perspectives like homotopy in Anandkumar et al. (2017), statistical physics (Arous et al. (2020), Ros et al. (2019), Wein et al. (2019) and Biroli et al. (2020)), quantum computing (Hastings (2020)) as well as statistical query (Dudeja & Hsu (2020)).

Recently, a fundamentally different set of mathematical tools that have been developed for tensors in the context of high energy physics have been used to approach the problem. They consist in trace invariants of degree $d \in \mathbb{N}$, obtained by contracting pair of indices of $d$ copies of the tensor $\mathbf{T}$. They have been used in Evnin (2020) to study the highest eigenvalue of a real symmetric Gaussian tensor. Subsequently, Gurau (2020) provided a theoretical study on a function based on an infinite sum of these invariants. Their results suggest a transition phase for the highest eigenvalue of a tensor for $\beta$ around $n^{1/2}$ in a similar way to the BBP transition in the matrix case (Baik et al. (2005)). Thus, this function allows the detection of a spike. However evaluating it involves computing an integral over a n-dimensional space, which may not be possible in a polynomial time.

The contribution of this paper is the use of these invariant tools to build tractable algorithms with polynomial complexity. In contrast to Gurau (2020), instead of using a sum of an infinite number of invariants, we select one trace invariant with convenient properties to build our algorithms. It lets us detect the presence of the signal linearly in time and with a space requirement in $O(1)$. Moreover, in order to recover the signal vector besides simply detecting it, we introduce new tools in the form of matrices associated to this specific invariant. Within this framework, we show as particular cases, that the two simpler graphs (of degree two) are similar to the tensor unfolding and the homotopy algorithms (which is equivalent to average gradient descent). These two algorithms are the main practical ones known from the point of view of space and time requirement (Anandkumar et al. (2017) provides a table comparison).

**Notations**    We use bold characters $\mathbf{T}, \boldsymbol{M}, \boldsymbol{v}$ for tensors, matrices and vectors and $T_{ijk}, M_{ij}, v_i$ for their components. $[p]$ denotes the set $\{1, \ldots, n\}$. A real $k$−th order tensor is of order $k$ if it is a member of the tensor product of $\mathbb{R}^{n_i}, i \in [k]$: $\mathbf{T} \in \bigotimes_{i=1}^{k} \mathbb{R}^{n_i}$ . It is symmetric if $T_{i_1 \ldots i_k} = T_{\tau(i_1) \ldots \tau(i_k)}$ $\forall \tau \in \mathfrak{S}_k$ where $\mathfrak{S}_k$ is the symmetric group (more details are provided in Appendix **??**). For a vector $\boldsymbol{v} \in \mathbb{R}^n$, we use $\boldsymbol{v}^{\otimes p} \equiv \boldsymbol{v} \otimes \boldsymbol{v} \otimes \cdots \otimes \boldsymbol{v} \in \bigotimes^p \mathbb{R}^n$ to denote its $p$-th tensor power. $\langle \boldsymbol{v}, \boldsymbol{w} \rangle$ denotes the scalar product of $\boldsymbol{v}$ and $\boldsymbol{w}$.

Let's define the operator norm, which is equivalent to the highest eigenvalue of a tensor of any order: $\|\mathbf{X}\|_{\mathrm{op}} \equiv \max \{\mathbf{X}_{i_1, \ldots, i_k} (\boldsymbol{w}_1)_{i_1} \ldots (\boldsymbol{w}_k)_{i_k}, \forall i \in \{1, \ldots, n\}, \|\boldsymbol{w}_i\| \leq 1\}$ The trace of $\boldsymbol{A}$ is denoted $\mathrm{Tr}(\boldsymbol{A})$. We denote the expectation of a variable $X$ by $\mathbb{E}(X)$ and its variance by $\sigma(X)$. We say that a function $f$ is negligible compared to a positive function $g$ and we write $f = o(g)$ if $\lim_{n \to \infty} f/g \to 0$.

**Einstein summation convention**    It is important to keep in mind throughout the paper that we will follow the Einstein summation convention: when an index variable appears twice in a single term and is not otherwise defined, it implies summation of that term over all the values of the index. For example: $T_{ijk} T_{ijk} \equiv \sum_{ijk} T_{ijk} T_{ijk}$. It is a common convention when addressing tensor problems that helps to make the equations more comprehensible.

## 2 GENERAL FRAMEWORK FOR SIGNAL DETECTION AND RECOVERY

### 2.1 WHAT DO WE USE TO STUDY THE SIGNAL?

An important concept in problems involving matrices is the spectral theory. It refers to the study of eigenvalues and eigenvectors of a matrix. It is of fundamental importance in many areas. In machine learning, the matrix PCA computes the eigenvectors and eigenvalues of the covariance matrix of the features to perform a dimensional reduction while ensuring most of the key information is maintained. In this case, the eigenvalues is a very efficient tool to describe data variability. In the case of signal processing, eigenvalue can contain information about the intensity of the signal, while the eigenvector points out to its direction. Lastly, a more theoretical example involves quantum physics where the spectrum of the matrix operator is used to calculate the energy levels and the state associated.

In all of these examples, an important property of the eigenvalues of a n-dimensional matrix $M$ is its invariance under orthogonal transformations $\{M \to OMO^{-1}, O \in \mathrm{O}(n)\}$ where $\mathrm{O}(n)$ is the $n$-dimensional orthogonal group (i.e. the group of real matrices that satisfies $OO^\top = I_n$, which should not be confused with the computational complexity $O(n)$). Since these transformations essentially just rotate the basis to define the coordinate system, they must not affect intrinsic information like data variability, signal intensity or the energy of a system. The eigenvalues are able to capture some of these inherent information, but recovering the complete general information requires computing their respective eigenvectors (for example to find the principal component, the direction of the signal or the physical state). There are more such invariants than eigenvalues. Another important set worth mentioning are the traces of the $n$ first matrix powers $\mathrm{Tr}(A), \mathrm{Tr}(A^2), \ldots, \mathrm{Tr}(A^n)$. Obtaining them uses slightly different methods than eigenvalues, but they contain the same information since each set can be inferred from the other through some basic algebraic operations.

On the basis of the matrix case, we expect that for a tensor $\mathsf{T} \in \bigotimes_{i=1}^{k} \mathbb{R}^{n_i}$, tensor quantities that are invariant under orthogonal transformations ($T_{a_j^1 \ldots a_j^k} \to O_{a_j^1 b_j^1}^{(1)} \ldots O_{a_j^k b_j^k}^{(k)} T_{b_j^1 \ldots b_j^k}$ for $O^{(i)} \in \mathrm{O}(n_i)$ $\forall i \in [k]$) should capture similar intrinsic information like the intensity of the signal, and conceivably, there should be other objects related to these quantities that are able to indicate the direction of the signal. However, the concept of eigenvalue and eigenvector is ill defined in the tensor case and not practical giving that the number of eigenvalues is exponential with the dimension $n$ (Qi (2005), Cartwright & Sturmfels (2013)) and computing them is very complicated. In contrast, we have a very convenient generalization of the traces of the power matrices for the tensors that we call trace invariants. They have been extensively studied during the last years in the context of high energy physics and many important properties have been proven (Gurau (2017)).

We first give a more formal definition of trace invariants. Let $\mathsf{T}$ be a tensor whose entries are $T_{i_1, \ldots, i_k}$. Let's define a contraction of a pair of indices as setting them equal to each other and summing over them, as in calculating the trace of a matrix ($A_{ij} \to \sum_{i=1}^{n} A_{ii}$). The trace invariants of the tensor $\mathsf{T}$ correspond to the different way to contract pairs of indices in a product of an even number of copies of $\mathsf{T}$. The degree of the trace invariants consists in the number of copies of $\mathsf{T}$ contracted. For example, $\sum_{i_1, i_2, i_3} T_{i_1 i_2 i_3} T_{i_1 i_2 i_3}$ and $\sum_{i_1, i_2, i_3} T_{i_1 i_2 i_2} T_{i_1 i_3 i_3}$ are trace invariants of degree 2. In the remainder of this paper, we will use the Einstein summation convention defined in the notation subsection.

A trace invariant of degree $d$ of a tensor $\mathsf{T}$ of order $k$ admits a practical graphical representation as an edge colored graph $\mathcal{G}$ obtained by following two steps: we first draw $d$ vertices representing the $d$ different copies of $\mathsf{T}$. The indices of each copy is represented by $k$ half-edges with a different color for each index position as shown in Figure 1a. Then, when two different indices are contracted in the tensor invariant, we connect their corresponding half-edges in $\mathcal{G}$. Reciprocally, to obtain the tensor invariant associated to a graph $\mathcal{G}$ with $d$ vertices, we take $d$ copies of $\mathsf{T}$ (one for each vertex), we associate a color for each index position, and we contract the indices of the $d$ copies of $\mathsf{T}$ following the coloring of the edges connecting the vertices. We denote this invariant $I_\mathcal{G}(\mathsf{T})$. Three important examples of trace invariants are: the melon diagram (Figure 1b) and the tadpole (1c). Avohou et al. (2020) provides a thorough study about the number of trace invariants for a given degree $d$. A very useful asset of these invariants is that we are able to compute their expectation for tensors whose components are Gaussian using simple combinatorial analysis (Gurau (2017)).

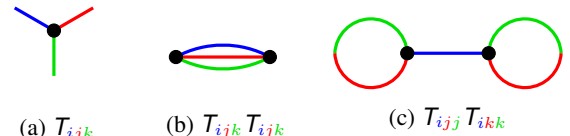

(a) $T_{ijk}$     (b) $T_{ijk} T_{ijk}$     (c) $T_{ijj} T_{ikk}$

Figure 1: Example of graphs and their associated invariants

## 2.2 WHAT DO WE USE TO RECOVER THE SIGNAL?

As previously mentioned in Section 2.1, an invariant should be able to detect a signal. But if our goal is to recover it, we should find mathematical objects that are able to provide a vector. To this effect, we introduce in this paper a new set of tools in the form of matrices. We denote by $M_{\mathcal{G},e}$ the matrix obtained by cutting an edge $e$ of a graph $\mathcal{G}$ in two half edges (see Figure 2 for an example). Indeed, this cut amounts to not summing over the two indices $i_1$ and $i_2$ associated to these two half-edges and using them to index the matrix instead. We will drop the index $\mathcal{G}, e$ of the matrix when the context is clear.

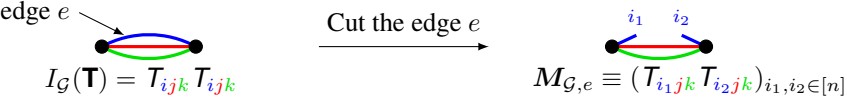

Figure 2: Obtaining a matrix by cutting the edge of a trace invariant graph $\mathcal{G}$

## 2.3 PHASE TRANSITION WITHIN THIS FRAMEWORK

We can represent the tensor from which we hope to extract the signal represented graphically as:

$$T_{ij_1\ldots j_{k-1}} \quad = \quad \beta\, v_i v_{j_1} \ldots v_{j_{k-1}} \quad + \quad Z_{ij_1\ldots j_{k-1}}$$

Figure 3: Graphical decomposition of the tensor $\mathbf{T}$

This decomposition leads us naturally to decompose in a similar way a tensor trace invariant $I_{\mathcal{G}}(\mathbf{T})$ into two parts, separating a first contribution $I_{\mathcal{G}}^{(N)}(\mathbf{T})$ associated to the pure noise tensor contribution and a second part $I_{\mathcal{G}}^{(S)}(\mathbf{T})$ enclosing all the other contributions, which are resulting from the addition of the signal.

$$I_{\mathcal{G}}(\mathbf{T}) = I_{\mathcal{G}}^{(N)}(\mathbf{T}) + I_{\mathcal{G}}^{(S)}(\mathbf{T}). \tag{2}$$

An identical decomposition can be carried out for the matrix. Let's consider a tensor $\mathbf{T}$, a graph $\mathcal{G}$ and its associated trace invariant $I_{\mathcal{G}}(\mathbf{T})$. Let's denote $I'_{\mathcal{G}}(\mathbf{T})$ the invariant associated to the subgraph obtained by removing from $\mathcal{G}$ the edge $e$ and its two vertices. We can distinguish three kind of contributions to the matrix $M_{\mathcal{G},e}$ that we denote $M_{\mathcal{G},e}^{(N)}, M_{\mathcal{G},e}^{(D)}$ and $M_{\mathcal{G},e}^{(R)}$, illustrated in Figure 4 (where we denoted the invariant $I'_{\mathcal{G}}(\mathbf{T})$ by $I'$ and dropped the index $\mathcal{G}, e$ for simplicity).

**Lemma 1.** $\mathbb{E}(M^{(N)}) = \frac{\mathbb{E}(I_{\mathcal{G}}^{(N)})}{n} I_n.$

Using the lemma 1, we identify three possible phases depending on which matrix operator norm is much larger than the others:

- **No detection and no recovery:** If $\left\| M^{(N)} - \mathbb{E}(M^{(N)}) \right\|_{op} \gg \left\| M^{(D)} \right\|_{op}, \left\| M^{(R)} \right\|_{op}$ then no recovery and no detection is possible we can't distinguish if there is a signal. It is for example the phase for $\beta \to 0$.

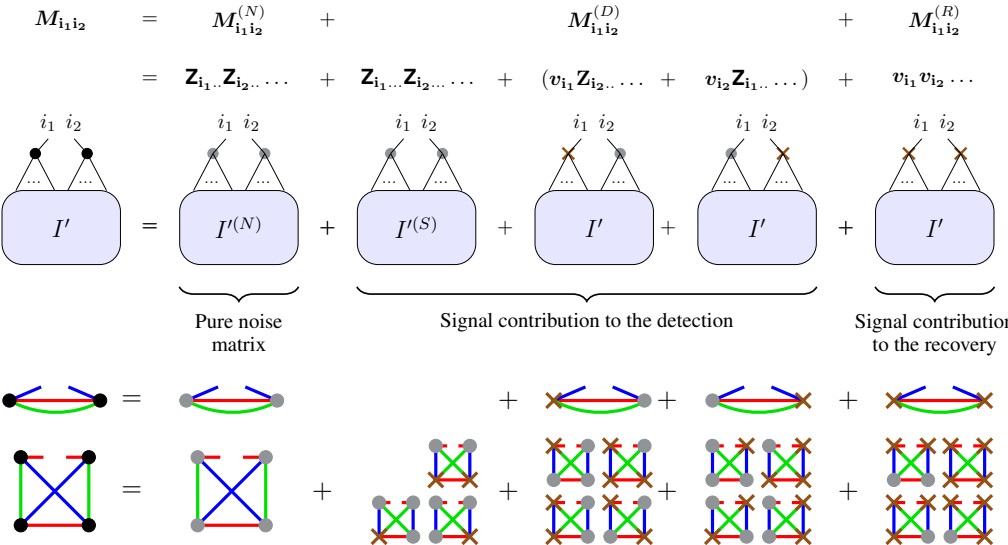

Figure 4: Decomposition of a matrix graph and the melon example

- **Detection but no recovery:** If $\left\|M^{(D)}\right\|_{\text{op}} \gg \left\|M^{(N)} - \mathbb{E}(M^{(N)})\right\|_{\text{op}}, \left\|M^{(R)}\right\|_{\text{op}}$ then detection but no recovery. We can detect the presence of the signal (thanks to the highest eigenvalue) but we can't recover the signal vector since the leading eigenvector is not correlated to the signal vector.

- **Detection and recovery:** $\left\|M^{(R)}\right\|_{\text{op}} \gg \left\|M^{(N)} - \mathbb{E}(M^{(N)})\right\|_{\text{op}}, \left\|M^{(D)}\right\|_{\text{op}}$. We recover the signal vector. It is for example the phase for $\beta \to \infty$.

## 2.4 ALGORITHMIC THRESHOLD FOR A GENERAL GRAPH

We can now state the important algorithms that will be essential for this paper. It is important to keep in mind that the following claims concern the large $n$ limit. Empirically, the approximation of large $n$ limit seems valid for $n > 25$.

---
**Algorithm 1:** Algorithm associated to the graph $\mathcal{G}$ and edge $e$

---
**Input:** The tensor $\mathsf{T} = \beta v^{\otimes k} + \mathsf{Z}$
**Goal:** Detection of $v$
**Result:** Gives the probability of the presence of a spike

---

The first algorithm gives a criteria for distinguishing a pure noise tensor from a tensor with a spike. Denoting $\mathbb{E}(I_{\mathcal{G}}(\mathbf{B}))$ the expectation and $\sigma(I_{\mathcal{G}}(\mathbf{B}))$ the variance of the trace invariant associated to a graph $\mathcal{G}$ for (**B**)), where the components of **B** are Gaussian random. The algorithm consists simply in calculating the trace invariant of the tensor and comparing its distance from $\mathbb{E}(I_{\mathcal{G}}(\mathbf{B}))$ with $\sigma(I_{\mathcal{G}}(\mathbf{B}))$. It is straightforward to see that calculating a trace invariant (which is a scalar) like $\mathsf{T}_{ijk}\mathsf{T}_{ijk}$ only needs $O(1)$ memory.

**Theorem 2.** *Let $\mathcal{G}$ be a graph of degree $d$, $\exists\ \beta_{det} > 0$ so that Algorithm 1 detects the presence of a signal for $\beta \geq \beta_{det}$.*

The second algorithm is able to recover the spike in a tensor **T** through the construction of the matrix of size $n \times n$ $M_{\mathcal{G},e}(\mathsf{T})$ associated to a given graph $\mathcal{G}$ and edge $e$.

---
**Algorithm 2:** Recovery algorithm associated to the graph $\mathcal{G}$ and edge $e$

---
**Input:** The tensor $\mathsf{T} = \beta v_0^{\otimes k} + \mathsf{Z}$
**Goal:** Estimate $v_0$
**Result:** Obtaining an estimated vector $v$

---

**Theorem 3.** *Let $\mathcal{G}$ be a graph of degree $d$, $\exists$ $\beta_{rec} > 0$ so that Algorithm 2 gives an estimator $\boldsymbol{v}$ so that $\boldsymbol{v}$ is strongly correlated to $\boldsymbol{v}_0$ ( $\langle \boldsymbol{v}, \boldsymbol{v}_0 \rangle > 0.9$) for $\beta \geq \beta_{rec}$.*

Since the algorithms 2 and 1 consists in algebraic operations on the tensors entries, they are very suitable for a parallel architecture. The Theorem 4 gives a lower bound to the threshold above which we can detect and recover a spike using a single graph. Interestingly, this threshold which appears naturally in our framework, matches the threshold below which there is no known algorithm that is able to recover the spike in polynomial time. We call the Gaussian variance of a graph $\mathcal{G}$, the variance of the invariant $I_{\mathcal{G}}(\mathbf{B})$ where $B_{ijk}$ are Gaussian random.

**Theorem 4.** *Let $k \geq 3$. It is impossible to detect or recover the signal using a single graph below the threshold $\beta \leq n^{\overline{k/4}}$ which is the minimal Gaussian variance of any graph $\mathcal{G}$.*

## 3  SOME APPLICATIONS OF THIS FRAMEWORK

Using these algorithms, we are now able to investigate the performance of our framework in various theoretical settings. In the first two subsections, we study the algorithms associated to two trace invariants of degree 2. They consist of the melonic diagram, which gives a very practical detection algorithm, and whose recovery algorithm is a variant of the unfolding algorithm, and the tadpole diagram whose recovery algorithm is similar to the homotopy algorithm. The last two sections are an illustration of the versatility of this framework. We study the case the dimensions $n_i$ of the tensor $\mathbf{T}$ ($\mathbf{T} \in \bigotimes_{i=1}^{k} \mathbb{R}^{n_i}$) are not necessarily equal, which is important for practical applications where the dimensions are naturally asymmetric. Our methods allows us to derive a new algorithmic threshold for this case.

### 3.1  THE MELON GRAPH SIMILAR TO TENSOR UNFOLDING

Let's consider the invariant $T_{i_1 \ldots i_n} T_{i_1 \ldots i_n}$ (illustrated by the graph in Figure 1b when $k = 3$). Its recovery algorithm (with the matrix obtained by cutting any of the edges) is similar to the tensor unfolding method presented in Richard & Montanari (2014). The difference is that the melonic algorithm uses only a matrix $n \times n$ instead of a matrix $n^{k/2} \times n^{k/2}$ for the tensor unfolding. However, the main contribution of this framework for this graph is that it allows the detection in a linear time ($n^k$ operations for a input (tensor) of size $n^3$ ) in a constant memory space (it just calculates a scalar). This provides it a potential usefulness as a first step for detecting the signal before deciding to use more computationally costly methods to recover it. Also, to the best of our knowledge, this framework is the first to theoretically prove a conjuncture that the unfolding algorithm works also for the symmetric case.

**Theorem 5.** *The algorithms 1 and 2 work for the melon graph with $\beta_{det} = \beta_{rec} = O(n^{k/4})$ in linear time and respectively $O(n^2)$ and $O(1)$ memory requirement.*

### 3.2  THE TADPOLE GRAPH

Figure 1c has a special characteristic: we can obtain two disconnected parts by cutting only one line. Therefore, the matrix obtained by cutting that edge is of rank one (in the form of $\boldsymbol{vv}^T$). Thus, the vector $\boldsymbol{v}$ has a weak correlation with the signal $\boldsymbol{v}_0$, which allow the tensor power iteration ($v_i \leftarrow T_{ijk} v_j v_k$) to empirically recover it (formal proofs require to consider some more sophisticated variants of power iteration like in Anandkumar et al. (2017) and Biroli et al. (2020)). This algorithm is a variant of the already existent homotopy algorithm.

**Theorem 6.** *The tadpole graph allows to recover the signal vector for $k \geq 3$ and $\beta = O(n^{k/4})$ by using local algorithms to enhance the signal contribution of the vector $T_{ijj}$.*

## 4  NUMERICAL EXPERIMENTS

In this section we will investigate the empirical results of the previously mentioned applications in order to see if they match with our theoretical results. We restrict to the dimension $k = 3$ for simplicity. More details about the experiments settings could be found in the Appendix.

## 4.1 COMPARISON OF RECOVERY METHODS

This distinction is easily visible, for $n = 100$ and $\beta = 100$ in Figure 5a where we plotted the histograms of the melonic invariant (in blue without signal and in orange with signal) for 500 independent instances of Gaussian random tensors **Z**. Thus, to measure the accuracy of the detection of the signal, we use the quantity: 1 - cardinal of the Intersection over the cardinal of the Union (1-IoU). Figure **??** suggests that a high probability detection requires $\beta \geq 3n^{3/4}$.

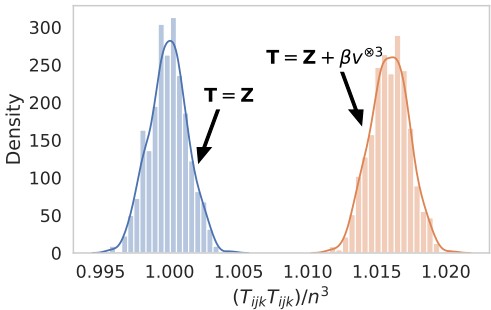

(a) Distribution of the melon invariant without (blue) and with (orange) signal.

Figure 5: Detection using the melonic graph.

For the recovery algorithm, we focus in the symmetric case (the most studied case and the most consistent with a symmetric spike) and, as in Richard & Montanari (2014), for every algorithm we use two variants: the simple algorithm outputting $v$ and an algorithm where we apply 100 power iterations on $v$: $v_i \leftarrow T_{ijk}v_jv_k$, distinguishable by a prefix "p-". In Figure **??**, we run 200 experiments for each value of $\beta$ and plot the $95\%$ confidence interval of the correlation of the vector recovered with the signal vector. We will compare our method to two type of results:

- Other algorithmic methods: the melonic (tensor unfolding) and the homotopy. To the best of our knowledge, they give the state of art respectively for the symmetric and asymmetric tensor (Biroli et al. (2020)). Other methods exist but are either too computationally expensive (sum of squares) or are variants of these algorithms.

- Information-theoretical results: In (Richard & Montanari (2014)), it was proven that computing the global minimum $v$ of the function $v \mapsto T_{ijk}v_iv_jv_k$ recovers the signal vector $v_0$ above a theoretical threshold $\beta_{\text{th}} = 2.87\sqrt{n}$ but with exponential time, and that no other approach can do significantly better than that. Thus, we plot in red dashed line denoted "perf" the deep minimum that is closest to $v_0$, by using gradient method with an initialization in $v_0$.

## 5 CONCLUSION

In this paper we introduced a novel framework for the tensor PCA based on trace invariants. Within this framework, we provide different algorithms to detect a signal vector or recover it. These algorithms use tensor contractions that has a high potential of parallelization and computing optimization. We illustrate the practical pertinence of our framework by presenting some examples of algorithms and prove their ability to detect and recover a signal vector linearly in time for $\beta$ above the optimal algorithmic threshold. Note that, one of the proposed detection algorithms requires only $O(1)$ memory requirement which could be advantageous in some applications. Moreover, we also show that two well known algorithms (Homotopy and Tensor Unfolding) can be mapped to our framework and result to simpler graph (e.g. the melonic graph). Important directions of future research is to apply these new methods to real data.

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
