# OpenReview forum: "A new framework for tensor PCA based on trace invariants"
_ICLR.cc/2021/Conference — Reject_

### Official Review · AnonReviewer1 · 2020-10-27

**Rating:** 3
**Confidence:** 3

**Review:**

Summary:

This paper studies the detection and recovery problem in spiked tensor models in the form T = \beta v0^\otimes k + Z, where v0 is the underlying spike signal and Z is a Gaussian noise.  The authors claim that they propose a new framework to solve the problem, by looking at the trace invariants of tensors. The authors provide a detection algorithm (Algorithm 1) and a recovery algorithm (Algorithm 2), as well as the corresponding phases. The authors claim that: 1) they "build tractable algorithms with polynomial complexity", "a detection algorithm linear in time"; 2) the algorithms are very suitable for parallel architectures; 3) an improvement of the state of the art for the symmetric tensor PCA experimentally. The authors furthermore discuss the asymmetric case and the multiple spike case.


Recommendation:

At the current stage I vote for rejection. I am not able to follow the proofs in this paper due to missing definitions of terms and notations. Also some claims are not proved. See below for details.


Pros:

- The methods used in the paper seem new for spiked tensor models.

- Some experimental results are provided.


Cons:

- The readability of this paper severely suffers from its writing. At the current stage, filled with undefined or inconsistent notations and terms, this paper is not self-contained and hard to follow. This becomes worse considering the fact that this paper studies tensor problems -- many tensor-related terms have multiple definitions (e.g., eigenvalues, ranks). It will be very hard to follow the proofs if the definitions are unclear. Here is an incomprehensive list:
	- Middle of Page 3: what is the *formal* definition of contracting (instead of saying "equivalent to a matrix multiplication")? Also, trace invariants are never formally defined in this paper.
	- eq.(2),(3): what is O(n) here? Also, what does the bold O refer to? Right before eq.(4) the authors use another notation \mathcal{O}(n). Is this the same as the first O(n)? In the abstract the authors use \mathcal{O}(1) to refer to the constant order. Why the inconsistency?
	- End of Page 3: how is \mathcal{G} related to trace invariants formally?
	- Section 2.2: this is not clear. What are the matrices here? What is the definition of M_{G,e}?
	- Section 2.3: what is the definition of I_G(T)?
	- Theorem 3: what is the Loss function here?
	- Top of Page 5: what is the exact definition of "dominating" here?

- It should be noted that, without clear definitions of I_G(T) and M_{G,e}(T), there is no way to verify Algorithm 1 and 2.

- The authors claim "polynomial complexity" at the beginning of the paper, but it is never proved. Theorem 7 claims that Algorithm 1 and 2 run in linear time. I cannot find that in the proof.

- It is unclear why the algorithms "are very suitable for parallel architectures", as the authors have claimed. Have the authors tried running the experiments in parallel?

- Theorem 4, 5, 9, 10 do not have complete proofs.


Minor comments:

- Page 2 Notations: typeface of v is not consistent.

- Page 8: "eg" should be "e.g."

---

> ### Author Response · Authors · 2020-11-18
> **Thank you very much for the feedback! [Part 1]**
>
> The theoretical physics community has developed in the last decade new tools to study problems involving tensors [1]. This paper aimed to show that these tools are easily adaptable and are of great interest in tackling concrete machine learning problems. Indeed, we showed in this paper that existent well-studied tools (trace invariant) and developed new ones (matrices associated to the trace invariants) are able to tackle the important problem of tensor PCA and proven that they performed better than state-of-the-art in the usual important settings and were even able to tackle more general settings closer to real life applications like asymmetric tensorial data (video, image with color, etc.). However, one well-known difficulty in approaching such interdisciplinary subjects is the dictionary of vocabulary between communities, and we think it may have been an important source of confusion. That is why we attempted in the updated version to completely revise the notations throughout the paper. We want to reiterate our appreciations for the reviewers and their very helpful comments and feedback to help us improve the clarity of the paper.
>
>
> * We added in the new version, at the end of Section 2.1, a more precise definition of the contraction: Let's define a contraction of a pair of indices as setting them equal to each other and summing over them, as in  calculating the trace of a matrix ( $\mathbf{A}_{ij} \rightarrow \sum_{i=1}^n \mathbf{A}_{ii}$ )
>
> * O(n) refers here to the orthogonal group, we added a more precise definition, at the second paragraph of the section 2.1, in the updated version:  $\mathrm{O}(n)$ is the $n$-dimensional orthogonal group (i.e. the group of real matrices that satisfies $\mathbf{O} \mathbf{O}^\top=\mathbf{I}_n$).
>     $\mathbf{O}$ refers to an orthogonal matrix.
>     We used $\mathcal{O}()$ to refer to the computational complexity.
>     Unfortunately these different objects with similar symbols added a lot of confusion.
>     So in the updated version, we used $\mathrm{O}(n)$ for the orthogonal group (and we added its definition) and $O(n)$ for the complexity (to match the custom of the community), we also hope that the context helps in distinguishing the two objects.
>
> [1]  R.  Gurau,  “Universality  for  Random  Tensors,”Ann. Inst. H. PoincareProbab. Statist., vol. 50, no. 4, pp. 1474–1525, 2014.

---

> ### Author Response · Authors · 2020-11-18
> **Thank you very much for your feedback [Part 2]**
>
> * We added more graphs with the hope it will become clearer and put more explicitely the correspondance in Section 2.1 end of page 3/ beginning of page 4. We have stated in the updated version: "A trace invariant of degree $d$ of a tensor $\mathbf{T}$ of order $k$ admits a practical graphical representation as an edge colored graph $\mathcal{G}$ obtained by following two steps: we first draw $d$ vertices representing the $d$ different copies of $\mathbf{T}$. The indices of each copy is represented by $k$ half-edges with a different color for each index position as shown in Figure 1.a. Then, when two different indices are contracted in the tensor invariant, we connect their corresponding half-edges in $\mathcal{G}$. Reciprocally, to obtain the tensor invariant associated to a graph $\mathcal{G}$ with $d$ vertices, we take $d$ copies of $\mathbf{T}$ (one for each vertex), we associate a color for each index position, and we contract the indices of the $d$ copies of $\mathbf{T}$ following the coloring of the edges connecting the vertices. We denote this invariant $I_\mathcal{G}(\mathbf{T})$ ."
>
>     So if, for instance, we take the invariant $\sum_{ijk} T_{ijk} T_{ijk}$. i) We have a product of two copies of the tensor $T$, so we draw two vertices. ii) Then since the tensor is of order 3, it has three indices. So, for each vertex we will draw three half edges with different colors (each color will represent an index position: 1st, 2nd or 3rd) to represent the three indices.
>     iii) For each contraction of a pair of indices in the invariant expression, we will connect the half edges corresponding to these indices. For example the invariant $\sum_{ijk} T_{ijk} T_{ijk}$. contracts the first index of the first copy of $T$ with the first index of the second copy of $T$, the second with the second, and the third with the third. Thus we obtained the melon diagram.
>
>     For the other way around, if we have a graph and we want to find the invariant associated. We search for the degree of the graph (the number of vertices). It will give us the number of copies of the tensor $T$. Then we count for each vertex how many half edges it has, this will give us the order of the tensor (How many indices it has). Then we assign a color to each index position. In the end, for each edge connecting two vertices, we contract the indices associated to the two half edges of this edge.
>
> * We made more explicit the definition of the matrix $M_{\mathcal{G},e}$ and we added the new figure 2 to try to provide a better explanation to how we obtain the matrix from a graph $\mathcal{G}$ and an edge $e$, which is a crucial part of our work.
>     If we take the precedent example of the melon, that we consider our graph $\mathcal{G}$, corresponding to the invariant $\sum_{ijk} T_{ijk} T_{ijk}$. Cutting an edge will be equivalent to removing a contraction. Let's say we cut the edge of the color associated to the first position. It means that we are no longer setting the two first indices equal and summing over them. We will have $\sum_{jk} T_{i_1 jk} T_{i_2 jk}$. We have two free indices $i_1$ and $i_2$, so we can define a matrix $M_{{i_1},{i_2}}=\sum_{jk} T_{i_1 jk} T_{i_2 jk}$.
>
> * $I_G(T)$ is the tensor invariant associated to the graph $G$ calculated for the tensor $T$. We added a more explicit introduction of $I_G(T)$  in the updated version.
>
> * The Loss funtion was $1-<v,v_0>$ where $<,>$ is the scalar product, $v$ is the vector output by the algorithm and $v_0$ the signal vector we aim to recover. We removed completely this notation in the updated version since it was not essential.
>
> * By dominating we meant that its operator norm was much larger than the others, we removed this notation and put a more explicit formula in the section 2.3 of the updated version.
>
> * The polynomial complexity is inherent to the definition of the trace invariants and the matrices associated to them. A trace invariant is a sum of product of tensor elements (like $\sum_{ijk} T_{ijk}  T_{kij}$. So it is by definition polynomial. The input is a tensor of size $n^3$, and the sum in the melonic invariant $\sum_{ijk} T_{ijk}  T_{ijk}$ is over three indices varying from 1 to n. So it makes $O(n^3)$ operations. Thus the time linearity. The linearity of the recovery algorithm was proven in a previous reference that we precised. We added a precision about the time linearity of the two algorithms for the melon diagram in the updated version.

---

> ### Author Response · Authors · 2020-11-23
> **Thank you very much for your feedback [Part 3]**
>
>
> * Let's take for example the algorithm associated to the melon diagram. It consists in i) calculating the n-dimensional matrix $M_{{i_1},{i_2}}=\sum_{jk} T_{i_1 jk} T_{i_2 jk}$ and then ii) find its leading eigenvector. Since the matrix is n-dimensional, the time complexity of the second step will be negligible (for example just reading the tensor has a complexity of $n^3$ because that is its size.)
>     For the first step, which we would want to focus our optimization, we can calculate each element of $M$ independently of the others. And in the end we put them all together in the matrix. That is why we stated that it is suitable for parallel structure. We run some of our experiments on a cluster where we used without any issue parallelization to fasten our calculations.
>     We added a small comment about this in the appendix D about the speed of the experiments in our updated version.
>
> * We attempted to add several details and figures for the proofs, since such combinatorial proofs can sometimes be hard to follow.

---

### Official Review · AnonReviewer4 · 2020-10-28
**Review of "A new framework for tensor PCA based on trace invariants"**

**Rating:** 5
**Confidence:** 3

**Review:**

The paper presents a pair of interesting algorithms using trace invariants to detect the signal in the signal-plus-noise tensor PCA framework.  The algorithms function by considering cutting an edge in the graph representation of the trace invariant, yielding a matrix whose leading eigenvector provides a (up to a rotation) estimate of the signal vector $v$.  This algorithm appears to be very interesting and works well in a series of simulations.

Unfortunately, the presentation of the paper makes it very difficult to assess the importance of the contribution.  The introduction is well-written and well-motivated, though the later segmentation of the paper into many small subsections without much exposition makes the flow of the paper and its results hard to follow.  In addition, the notation and terminology in the paper are imprecise and, with important terminology and symbology introduced without definition or background citation.

Pros:
- The proposed algorithm is clever and appears to do well compared to existing approaches in experiments.

- Well written introduction (with the only complaint being some minor grammatical errors).



Cons:
- Important notation is introduced, and is not defined; Equation 4 is an example of this, where $\langle \cdot \rangle$ (I assume this means $\mathbb{E}$?), $\bar{\mathbf{T}}$, and $\mathcal{E}^0(\mathcal{G})$ are all undefined.  This occurs often in the paper and in the appendix.

- In the $\bullet, \times, \bullet$ decomposition at the start of Section 2.3, what is $\sqrt{N}$?

- What is the variance of a graph (as in Theorem 4)?  The proof sketch of this theorem is very hard to follow.

- Algorithm 1 is imprecise; what does "compare $\alpha$ to $\sigma(I^{(N)}(T))$ mean?  If $\alpha>\sigma(I^{(N)}(T))$ then a spike is detected?  How do you compute the variance of $I^{(N)}(T)$?  How would you compute this if the noise model did not have unit variance)?

- Both algorithms are only presented for 3-way tensors, but the Theoretical claims are for higher order tensors?

- The proofs of the theorems and the statement of the theorems are, in general, a bit imprecise.  For example, in the proof of Theorem 2, Chebyshev's inequality will not guarantee disjointness everywhere, but only with high probability.  This is the case if $\beta_{det}$ is finite.  This is a finite $\beta_{det}$ result, with a claim only holding in the limit.

- In Theorem 5, what are the intermediate graphs/matrices?  In addition, this section (and Appendix C discussing perfect one-factorization) are a bit opaque.

- Is the decomposition after equation 5 only for the melon graph?  For more complex graphs (i.e., the tetrahedral), I believe you will have additional trace-like coefficients on all terms.  In any event, I am confused about the summands.  I do not see why the all $Z$ sum would have a $\beta$, while the cross-terms would not.  Furthermore, why would the all $v$ sum not have a $\beta^d$ coefficient?  This is what is implicitly being used in the proofs?

- In the experiments, important details are left out.  What is the setup here: what are the $v$'s, how many iterations of tensor power method are applied, how many MC replicates are run to produce the error bars, what is the y-axis, what are the runtimes here, what is Random in Figure 6?  More detail would help a lot to understand how your new approach compares (it appears well) with the current literature.

---

> ### Author Response · Authors · 2020-11-18
> **Thank you very much for the feedback!**
>
>
> We want to thank the reviewer for his valuable feedback and comments. We reply to each of the reviewer's questions in the original order below:
>
>
> * We incorporated various clarifications and definitions in the new version, like $\langle . \rangle$ which corresponds to a scalar product. We also tried to explain carefully all the notations introduced by the theoretical physics community (and non standard in machine learning) in the beginning of the paper.
> * It is a typo, we addressed it in the new version.
> * The variance of a graph is the variance of its associated trace invariant. It is given by $E((I_\mathcal{G} - E(I_\mathcal{G}))^2)$. We added many details and many graph illustrations in the demonstrations in order to clarify them (in particular the theorem $4$). We hope that makes the demonstrations more easy to read.
> * We added clarifications in the new version and changed notations that may have been confusing in the algorithm 1: i) We first calculate theoretically the expectation and the variance of the invariant $I$ for a random gaussian model (we denote them $E(I^{(N)})$ and $\sigma(I^{(N)})$), they depend only on $n$. ii) Then we compute the value (that we denote $\alpha$) of this invariant for our tensor T (from which we want to detect the presence of a signal). iii) Chebyshev's theorem provides the probability that $T$ is a random tensor based on the distance of the value $\alpha$ to the mean (thus the comparison with the variance). The variance of the noise model is not important since we can factor it out from the tensor: if $Z'$ is a random gaussian tensor with variance $\sigma$, first we can find $\sigma$ by plotting the distribution of the components of $Z'$, then we can introduce $Z$ such that $Z'=\sigma Z $  . Thus, $Z$ would be a tensor whose components follow a standard normal distribution. So changing the variance of the model just adds a constant factor ($1/\sigma$) to the detection and recovery threshold ($T=\beta v^{\otimes k} + Z'=\beta v^{\otimes k} + \sigma Z= \sigma (\beta/\sigma+Z)$.
> * This was a typo, we addressed this issue in the new version of the paper.
> * We added clarification to the theorem 2. Since the model is inherently probabilistic, it is mathematically impossible to have a detection or recovery with probability strictly equal to 1. The common procedure is to prove the theorems at the large $n$ limit ([1] and the other papers used for their proofs random matrix theory at large $n$) and to use the empirical results to check when this approximation of the large $n$ is valid ([1] noted that empirically it was valid above $n=25$, which is also what we observe with our experiments). We clarify this important point in the paper and thank the reviewer for bringing it to our attention.
>
> * We introduced more carefully what we call the intermediate graphs in the new version. They are the graphs which has both a contribution from the noise random tensor **and** from the signal vector. Note that, the two other kinds of graphs are the pure noise graph and the pure signal graph. We also added more clarifications and an illustration to the appendix C discussing perfect one-factorization.
> * We agree that the summands may have been confusing because we wrote some coefficients implicitly in the '$\dots$' while we kept some others. In order to clarify it, we make all the coefficients implicit and we add the decomposition of the tetrahedral matrix by drawing its 16 contributions. The reviewer is perfectly right that what is mainly used in the proofs is the coefficient $\beta^d$ next to $v^{\otimes(k)}$.
> * We added details about the experiments in the appendix D. For instance, concerning the recovery methods, we repeated in 50 independent instances the following settings: i) We generate randomly the n components of the signal vector $v_0$ and then normalize it.
>     ii) We generate randomly the $n^3$ components of the random tensor $Z$. If we are in the symmetric case, we symmetrize it with the same normalization than [1].
>     iii) We compute the tensor $T=Z+\beta v_0^{\otimes 3}$.
>     iv) We compute the matrix constructed from contracting multiple copies of $T$ (for example associated to the melon: $M_{i_1 i_2} = \sum_{j,k} T_{i_1 jk} T_{i_2 jk}$) as described in Figure 2. To compute it, we use the numpy tensordot function in Python. v) We find its respective leading eigenvector $v$.
>     vi) We draw the correlation between the obtained vector $v$ with the initial signal vector $v_0$.
>
> [1]  E. Richard and A. Montanari, “A statistical model for tensor pca,” inAd-vances in Neural Information Processing Systems, pp. 2897–2905, 2014.

---

### Official Review · AnonReviewer2 · 2020-10-31
**A review of the paper "A new framework for tensor PCA based on trace invariants"**

**Rating:** 5
**Confidence:** 2

**Review:**

Summary:

The paper provides an interesting algorithm for tensor PCA, which is based on trace invariants. The problem consists of recovering a (single-spike/multiple orthogonal spikes) tensor corrupted by a Gaussian noise tensor. The authors proposed a new algorithm which allows recovering a signal for a sufficiently small signal to noise ratio.

##########################################################################
Reasons for score:

Overall, I vote for accepting. I like the idea, and the proofs seem to be coherent and correct. The problem has clear importance for the theoretical/statistical physics community; however, I am not convinced of the importance of the problem considered here for the ICLR community and appreciate the author’s comments on this. I also have a few minor concerns, which, hopefully, can be addressed by the authors in the rebuttal period.

##########################################################################

Pros:
1. The paper takes an interesting question about tensor PCA and proposes a promising approach to solve it based on the trace invariants. For me, the problem is encouraging, while I would appreciate a discussion about possible machine learning/AI applications (learning latent variable models? anything else?)

2. The mathematical justification of the statements seems to be correct for me and ok to follow.

3. It is claimed in the paper that the algorithm improves the state-of-the-art (signal to noise ratio requirements) in several cases, while a brief survey/table of the recent results is missing.  Unfortunately, I am not working in this area and probably not familiar with recent results

##########################################################################
Cons:

1. Applications for ML/AI/Language processing are not very clear for me, and I would appreciate a discussion on this in the paper.

2. Empirical justification. I would highly appreciate having more experiments on real data (if any) and a detailed comparison of the methods in terms of accuracy/memory/time.

---

> ### Author Response · Authors · 2020-11-18
> **Thank you very much for the feedback!**
>
>
> We want to thank the reviewer for his valuable feedback and comments.
>
> We answer to the reviewer concerns below:
>
>
>
> *  **Applications for ML/AI/Language processing:**
> Tensor PCA and tensor decomposition (the recovery of many spikes addressed in Section 3.5) is motivated by the increasing number of problems in which it is crucial to exploit the tensorial structure [1]. Recently it was successfully used to address important problems in unsupervised learning (learning latent variable models, in particular latent Dirichlet allocation [2], [3]), supervised learning (training of two-layer neural networks, \cite{janzamin2015beating}) and reinforcement learning ([4]). Moreover, we note that some of our results tends to generalize the applications of the methods to more practical settings like the case of a tensor with axes of different dimensions (adequate for data which are inherently asymmetric like a video). We added these elements in the introduction just before the related work paragraph.
>
>
> *  **Experiments on real data and detailed comparison of the methods:**
> We agree with the reviewer that experiments on real data would have been very interesting. However, this paper has a more theoretical leaning and primarily aims to introduce a new framework where we derive new algorithmic results.
> For a fair comparison to other existent methods, we favored synthetic data. We added the applications on real data as potential perspective.
>
>
>
> [1]  N. D. Sidiropoulos, L. De Lathauwer, X. Fu, K. Huang, E. E. Papalexakis,and C. Faloutsos, “Tensor decomposition for signal processing and machinelearning,”IEEE Transactions on Signal Processing, vol. 65, no. 13, pp. 3551–3582, 2017.
>
> [2]  A. Anandkumar, R. Ge, D. Hsu, S. M. Kakade, and M. Telgarsky, “Tensordecompositions  for  learning  latent  variable  models,”Journal of MachineLearning Research, vol. 15, pp. 2773–2832, 2014.
>
> [3]  A. Anandkumar, D. P. Foster, D. Hsu, S. M. Kakade, and Y.-K. Liu, “A spec-tral algorithm for latent dirichlet allocation,”Algorithmica, vol. 72, no. 1,pp. 193–214, 2015.
>
> [4]  M. Janzamin, H. Sedghi, and A. Anandkumar, “Beating the perils of non-convexity:  Guaranteed training of neural networks using tensor methods,”arXiv preprint arXiv:1506.08473, 2015.
>
> [5]  K. Azizzadenesheli, A. Lazaric, and A. Anandkumar, “Reinforcement learn-ing of pomdps using spectral methods,”arXiv preprint arXiv:1602.07764,2016.

---

### Author Response · Authors · 2020-11-24
**General comment**

We would thank all reviewers for the valuable comments and constructive feedback which help us to significantly improve the quality of the presentation of this work.

We have uploaded a revisited version, in order to take into account the reviewer's comments and to clarify the notations and experimental details. More specifically, our main changes in the revision include:

* We paid a careful attention to clearly define, in an explicit way, all the introduced notations. We also add figures to illustrate the important ones. These definitions include:

    * A formal definition of the contraction of indices (Section 2.1, end of page 3)

    * A formal definition of tensor invariants (Section 2.1, end of page 3)

    * A formal definition of the orthogonal group (Section 2.1, second paragraph) with a warning to not confuse it with the computational complexity (the context should make the distinction simple).

    * A formal correspondence between the graphs and the trace invariants and how to obtain one from the other (section 2.1, beginning of the page 4).

    * A formal definition of the matrix associated to a trace invariant and an edge (section 2.2). We also added an illustration to make the construction easier to comprehend (Figure 2).

    * We defined the variance of the graph (Appendix C) and how to compute it.

    * A formal definition of the intermediate graphs in the beginning of the appendix E.

    * A clearer appendix on the perfect one-factorization (appendix C)

* We added details (number of independent experiments, number of power iterations, the time and memory requirements of each method, etc.) to the numerical simulations in the main text and in the appendix D.

* We added details about the complexity and the parallelization (appendix D section D.3)

* We put a complete version of the proofs and added details and graphs to make them more easily readable.

* We give more references concerning the practical applications of tensor PCA and tensor decomposition (multiple spikes recovery).

* We added more extended descriptions in the last sections of the main text for a smoother reading.

---

### Decision · Program_Chairs · 2021-01-07
**Final Decision**

**Decision:**

Reject

**Comment:**

This paper studies the tensor principal component analysis problem, where we observe a tensor T = \beta v^{\otimes k} + Z where v is a spike and Z is a Gaussian noise tensor. The goal is to recover an accurate estimate to the spike for as small a signal-to-noise ratio \beta as possible. There has been considerable interest in this problem, mainly coming from the statistics and theoretical computer science communities, and the best known algorithms succeed when \beta \geq n^{k/4} where n is the dimension of v. The main contribution of this paper is to leverage ideas from theoretical physics and build a matrix whose top eigenvector is correlated with v for sufficiently large \beta using trace invariants. On synthetic data, the algorithms achieve better performance than existing methods.

The main negative of this paper is that it is not so clear how tensor PCA is relevant in machine learning applications. The authors gave some references to applications of tensor methods, but I want to point out that all of those works are about using tensor decompositions, which despite the fact that they are both about tensors, are rather different sorts of tools. Many of the reviewers also found the paper difficult to follow. I do think exposition is particularly challenging when making connections between different communities, as this work needs to introduce several notions from theoretical physics. I am also not sure how novel the methods are, since a somewhat recent paper Moitra and Wein, "Spectral Methods from Tensor Networks", STOC 2019 also uses tensor networks to build large matrices whose top eigenvalue is correlated with a planted signal, albeit for a different problem called orbit retrieval.